# Do outcomes after kidney transplantation differ for black patients in England versus New York State? A comparative, population-cohort analysis

Sanna Tahir,[1] Holly Gillott,[1] Francesca Jackson-Spence,[1] Jay Nath,[1] Jemma Mytton,[1] Felicity Evison,[2] Adnan Sharif[1,3]

► Prepublication history and additional material is available. To view please visit the journal (http://dx.doi.org/ 10.1136/bmjopen-2016-014069).

[1]College of Medical and Dental Sciences, University of Birmingham, Birmingham, UK
[2]Department of Informatics, University Hospitals Birmingham NHS Foundation Trust, Birmingham, UK
[3]Department of Nephrology and Transplantation, Queen Elizabeth Hospital, Birmingham, UK

**Correspondence to**
Dr. Adnan Sharif;
adnan.sharif@uhb.nhs.uk

## ABSTRACT

**Objectives** Inferior outcomes for black kidney transplant recipients in the USA may not be generalisable elsewhere. In this population cohort analysis, we compared outcomes for black kidney transplant patients in England versus New York State.

**Design** Retrospective, comparative, population cohort study utilising administrative data registries.

**Settings and participants** English data were derived from Hospital Episode Statistics, while New York State data were derived from Statewide Planning and Research Cooperative System. All adults receiving their first kidney-alone allograft between 2003 and 2013 were eligible for inclusion.

**Measures** The primary outcome measure was mortality post kidney transplantation (including inhospital death, 30-day mortality and 1-year mortality). Secondary outcome measures included postoperative admission length of stay, risk of rehospitalisation, development of cardiac events, stroke, cancer or fracture and finally transplant rejection/failure. Cox proportional hazards regression was used to investigate relationship between ethnicity, country and outcome.

**Results** Black patients comprised 6.5% of the English cohort (n=1215/18 493) and 23.0% of the New York State cohort (n=2660/11 602). Compared with New York State, black kidney transplant recipients in England were more likely younger, male, living-donor kidney recipients and had dissimilar medical comorbidities. Inpatient mortality was not statistically different, but death within 30 days, 1 year or kidney transplant rejection/failure was lower among black patients in England versus black patients in New York State. In adjusted regression analysis, with black ethnicity the reference group, white patients had reduced risk for 30-day mortality (OR 0.62 (95% CI 0.44 to 0.86)) and 1-year mortality (OR 0.79 (95% CI 0.63 to 0.99)) in New York State but no difference was observed in England. Compared with England, black kidney transplant patients in New York State had increased HR for kidney transplant rejection rejection/failure by median follow-up (HR 2.15, 95% CI 1.91 to 2.43).

**Conclusions** Outcomes after kidney transplantation for black patients may not be translatable between countries.

## INTRODUCTION

Black patients are significantly more likely to develop end-stage kidney disease (ESKD), which is likely secondary to increased risk of chronic kidney disease progressing to kidney failure.[1] Adjusted incident rates for black patients (or African-Americans) in the United States Renal Data System (USRDS) in 2014 (latest available data) was the highest for any ethnicity group, with a ESKD incidence rate ratio versus white patients of 3.1.[2] Even in countries with a lower incidence of ESKD than the USA like the UK, there is a disproportionate number of black patients with ESKD undergoing renal replacement therapy compared with the general population.[3] Therefore, strategies to optimise care for black patients with ESKD remains of critical importance.

While kidney transplantation is recognised as the preferred modality of renal replacement therapy for patients with ESKD, disparate post-transplant outcomes are reported for black recipients of a kidney allograft. For example, black kidney transplant patients in the USA have previously been shown to have worse kidney allograft survival compared with white recipients,[4] although recent analyses suggest reduced disparity (but still not equality) in kidney allograft outcomes between black and white patients in the contemporary era in the USA.[5] However, this is not generalisable across North America as Canadian black transplant patients appear to have similar kidney allograft survival and improved mortality compared with Canadian white kidney transplant patients.[6] Mortality

differences after kidney transplantation between the USA and Canada are well recognised, with magnification of risk for death observed in black Americans versus Canadians.[7] In a European setting, Pallet and colleagues have shown no difference in outcomes for black versus white patients after kidney transplantation in France,[8] and no mortality difference is observed between black and white patients in the UK.[9 10] Therefore, inferior mortality observed for black kidney transplant patients widely attributed to the USA may be country specific, thus making interpretation of such data difficult for other population cohorts like the UK.

**Strengths and limitations of this study**

► Large population-cohort study of two datasets.
► Data linkage to mortality data for accuracy.
► Use of administration rather than registry data.
► Kidney allograft survival data not well defined.
► Numerous confounders unavailable to robustly probe this question.

No comparative analysis has ever been conducted to specifically compare post-transplant outcomes for black kidney transplant patients between countries across distinct geographical regions like the UK and the USA. This is of interest as healthcare systems differ significantly between the two countries, with the possibility that black people in the UK may not share the same post-transplant outcomes observed in the USA. Therefore, it is unclear if the counselling of black kidney transplant candidates in one country can be adequately made utilising data translated from black patients in another country. The aim of this analysis was to undertake a comparative population-based cohort analysis exploring outcomes after kidney transplantation for black patients in England versus New York State.

## MATERIALS AND METHODS

### Study population

We obtained data from every patient's first kidney-alone transplant procedure performed in England and New York State between 2003 and 2013, collecting patient demographics that included age, gender, donor type (living or deceased), transplant year, medical comorbidities (based on International Classification of Disease, 10th revision (ICD-10) codes) and ethnicity. English data were obtained from Hospital Episode Statistics,[11] an administrative data warehouse containing admissions to all National Health Service (NHS) hospitals in England. It contains detailed records relating to individual patient treatments, with data extraction facilitated utilising codes on procedural classifications (Office of Population Censuses and Surveys Classification of Interventions and Procedures, fourth revision (OPCS-4))[12] and medical classifications (WHO ICD-10).[13] The comparative analysis with the USA was performed with contemporaneous New York State data and extracted from the Statewide Planning and Research Cooperative System (SPARCS), a comprehensive all-payer data reporting system collecting patient level data across New York State.[14] The database collects information including patient demographics, diagnoses and procedures and charges for every inpatient hospital admission, ambulatory surgical procedure, and emergency department admission. Individuals are assigned a unique, encrypted identification code, allowing for longitudinal analyses. Estimated reporting

completeness obtained from SPARCS inpatient annual reports during the study period (2000–2011) ranged from 95% to 100%, with an average of more than 98%.

This study included all patients' first kidney transplant procedures (OPCS-4 codes; M01(0–5,8,9)) performed in England and New York State between the years 2003–2013. With regards to outcome analysis, both Hospital Episode Statistics (HES) and SPARCS data sets have the limitation of only capturing deaths occurring in a hospital setting. To obtain the complete mortality list, the study cohort was cross-referenced with mortality data from the Office for National Statistics and New York State/New York City Vital Statistics, respectively, which collects information on all registered deaths in the UK and New York State, respectively. Death data were available from 1 January 2013 up until 31 December 2014. Combining sources via this data linkage process creates a comprehensive data set with regards to mortality, which was the endpoint of interest in this analysis. In addition, we extracted data for kidney transplant rejection/failure (ICD-10: T861 or ICD-9: 99 681), which is derived from the administrative data and substituted for lack of data linkage to transplant registry records. Formal ethical approval was not required due to the pseudonymised nature of the data retrieved; data were linked by NHS Informatics utilising a special HES ID code and avoided any patient identifiable information. Thus, the study was registered as an audit with University Hospitals Birmingham NHS Trust (audit identifier; CARMS-12578).

### Data inclusion

We extracted data on all adult kidney allograft recipients' first procedure between the dates of 1 January 2003 and 31 December 2013, who underwent their kidney transplant procedure in a transplant centre in either England or New York State. From our original cohorts of 21 371 and 12 373 patients in England and New York States, respectively, we excluded from analysis 2582 and 771 patients from both data sets, respectively, as shown in online supplementary figure 1. Ethnicity status was missing in <5 cases (too small to identify), although anyone classified as unknown was included under 'other' for analysis. All patients were followed up until December 2014.

### Outcomes

The primary outcome measure was all-cause mortality post kidney transplantation (including inhospital death, 30-day mortality and 1-year mortality). In addition, we looked at various secondary outcome measures including postoperative admission length of stay, risk of re-hospitalisation, development of cardiac events (ICD-10: I2(01234), ICD-9: 410,411,413,429), stroke (ICD-10: I6(0 1 2 3 4 5 6), ICD-9: 430–436), cancer (ICD-10:C00-C99, ICD9: 140–199) or fracture (ICD-10: S(0123456789)2, T02, T07, T10, T12, ICD-9: 800–829) and finally kidney transplant rejection/failure (ICD-10: T861, ICD-9: 99 681). Rehospitalisation was defined as whether a patient was readmitted to hospital as an emergency at any time postdischarge of

their transplant admission. All other secondary outcomes were captured with use of ICD-10 and ICD-9 codes (as defined above), at any time post-transplant.

## Statistical analysis

Differences between groups were compared using $\chi^2$ tests for categorical variables two-sample t-test for normal continuous variables or Mann-Whitney tests for all non-normal continuous variables. Multivariate logistic regression was used to measure 30-day and 12-month mortality. All patients were included in these models to see if there was a difference in mortality outcomes for the different ethnic groups, within the two populations. Variables included in the models were age, sex, admission method (emergency vs elective), number of emergency readmissions and ethnic group. Survival analyses were performed, combining all black patients in the two countries, for the outcome of graft failure/rejection (defined using ICD-10: T861 or ICD-9: 99681 at any time post-transplant) using Cox's proportional hazards model and the generation of Kaplan-Meier plots. The proportional hazard assumption was checked and satisfied by examination of the scaled Schoenfeld residual plots. Variables included in the Cox model were ethnicity, age, gender, donor type (living vs deceased), year of transplant, geography (England vs New York State) and selected medical comorbidities at the time of transplant (history of myocardial infarction, peripheral vascular disease, cerebrovascular disease, congestive cardiac failure and diabetes). Sensitivity analyses, stratified by ethnicity and donor type, were further conducted for both English and

New York State. A p value less than 0.05 was considered statistically significant in the analysis. All statistical analyses were conducted using Stata V.14.

## RESULTS

Between 2003 and 2013, there were 21371 patients in England and 12373 patients in New York State who had their first kidney transplant. After exclusions due to missing age or sex, and those who had multiorgan transplants, data were analysed for 18493 and 11602 patients between 2003 and 2013 from England and New York State, respectively, with median follow-up in England and New York State of 6.3 years and 5.5 years, respectively. Black patients comprised 6.5% of the English cohort (n=1215) and 23.0% of the New York State cohort (n=2660), forming the basis of all comparative analyses.

## Baseline demographics

Table 1 compares demographics between black patients in England versus New York State. Black kidney allograft recipients in England versus New York State were found to be younger (46.1 vs 50.1, respectively, p<0.001), more likely to be male (61.2% vs 56.8%, respectively, p=0.011) and more likely to get living-donor kidneys (28.2% vs 20.6%, respectively, p<0.001). We also observed black recipients in England versus New York State had less medical comorbidities listed at time of transplantation including diabetes (16.0% vs 27.9%, respectively, p<0.001), previous myocardial infarcts (2.1% vs 3.7%, respectively, p<0.001), strokes (1.2% vs 1.7%, respectively,

| Table 1 | Baseline demographics of black kidney transplant recipients in England versus NYS between 2003 and 2013* | | | |
|---|---|---|---|---|
| **Total** | | **England (n=1215)** | **NYS (n=2660)** | **p Value** |
| Age | Mean±SD | 46.3±12.4 | 50.1±13.5 | <0.001* |
| Total length of stay | Median±IQR | 10±8–16 | 6±5–9 | <0.001† |
| Post-transplant hospital stay | Median±IQR | 8±6–13 | 6±4–8 | <0.001† |
| Sex, n (%) | Male | 743 (61.2) | 1511 (56.8) | 0.011‡ |
| | Female | 472 (38.8) | 1149 (43.2) | |
| Type of donor, n (%) | Living | 343 (28.2) | 548 (20.6) | <0.001‡ |
| | Deceased | 856 (70.5) | 1555 (58.5) | |
| | Unknown | 16 (1.3) | 557 (20.9) | |
| Medical comorbidities at time of transplantation, n (%) | Diabetes | 194 (16.0) | 741 (27.9) | <0.001‡ |
| | Acute MI | 25 (2.1) | 99 (3.7) | |
| | Cerebrovascular disease | 14 (1.2) | 46 (1.7) | |
| | Congestive heart failure | 22 (1.8) | 220 (8.3) | |
| | Liver disease | 6 (0.5) | 16 (0.6) | |
| | Peptic ulcer | 8 (0.7) | 31 (1.2) | |
| | Peripheral vascular disease | 13 (1.1) | 83 (3.1) | |

*t-Test.
†Mann-Whitney test.
‡$\chi^2$ test.
NYS, New York State.

Table 2  Comparing 1 year and follow-up outcomes after kidney transplantation for black patients in England versus New York State between 2003 and 2014

| Variable | England, n (%) | New York State, n (%) | p Value |
|---|---|---|---|
| 1-Year follow-up | | | |
| Emergency readmission | 782 (64.36) | 892 (33.53) | <0.001 |
| Cardiac event | 16 (1.32) | 79 (2.97) | 0.002 |
| Stroke | 12 (0.99) | 23 (0.86) | <0.001 |
| Cancer | 17 (1.40) | 26 (0.98) | 0.245 |
| Fracture | 5 (0.41) | 18 (0.68) | 0.319 |
| Transplant rejection/failure | 226 (18.6) | 702 (26.4) | <0.001 |
| Any time post-transplant | | | |
| Number of emergency readmissions (mean) | 4.33 | 3.99 | 0.0697 |
| Cardiac event | 72 (5.93) | 336 (12.63) | <0.001 |
| Stroke | 47 (3.87) | 125 (4.70) | 0.224 |
| Cancer | 71 (5.84) | 171 (6.43) | 0.485 |
| Fracture | 36 (2.96) | 84 (3.16) | 0.745 |
| Transplant rejection/failure | 367 (30.2) | 1590 (59.8) | <0.001 |
| Mortality | | | |
| Inhospital death | 9 (0.74) | 36 (1.35) | 0.099 |
| 30-day mortality | 12 (0.99) | 61 (2.29) | 0.006 |
| 12-month mortality | 34 (2.80) | 173 (6.50) | <0.001 |

Obtained using $\chi^2$ tests.

p<0.001) and congestive cardiac failure (1.8% vs 8.0%, respectively, p<0.001), suggesting a higher level of medical comorbidity listed at the time of kidney transplantation for black patients in New York State versus England.

### Post-transplant outcomes

Table 2 demonstrates the difference in mortality outcomes comparing black kidney transplant patients in England versus New York State. While inpatient mortality was similar comparing black patients in England versus New York State (0.7% vs 1.4%, respectively, p=0.099), mortality was significantly lower among black patients in England versus New York State within 30 days (1.0% vs 2.3%, respectively, p=0.006) and 1 year (2.8% vs 6.5%, respectively, p<0.001)) after kidney transplantation.

There were significant differences in the post-transplantation course for black kidney transplant patients in England versus New York State (see tables 1 and 2). Black patients in England versus New York State had longer postoperative inpatient stays (median 8 vs 6 days, p<0.001) and had a greater tendency to be readmitted within the first year post-transplant (64.36% vs 33.53%, p<0.001), but overall mean number of emergency rehospitalisation episodes post-transplant by median follow-up was similar between the two cohorts (4.3 vs 4.0, respectively, p=0.07). With regards to medical events occurring post kidney transplantation, black patients in England versus New York State were less likely to have cardiac events (6.0% vs 12.6%, respectively, p<0.001), but no difference was observed in episodes of cancer, strokes or

fractures after kidney transplantation for black patients in England versus New York State.

### Regression analysis

While our analyses revealed significant mortality differences between black patients in England versus New York State, we were keen to understand if black ethnicity itself was a risk factor for mortality in each population cohort. In a logistic regression model, accounting for variables associated with mortality after kidney transplantation (age, sex, admission method and number of emergency readmissions), we analysed whether black ethnicity was an independent predictor for death in England or New York State. With black ethnicity utilised as the reference group, table 3 displays the output from the logistic regression analysis and shows white kidney allograft recipients had reduced risk for 30-day mortality (OR 0.62 (95% CI 0.44 to 0.86)) and 1-year mortality (OR 0.66 (95% CI 0.55 to 0.81)) in New York State, but no difference was observed in England (30-day mortality: OR 0.81 (0.44 to 1.49) and 1-year mortality: OR 0.94 (0.65 to 1.34). Additional sensitivity analyses exploring risk for mortality between both cohorts, stratified by ethnicity and donor type, were conducted and are shown in the online supplementary material.

A significant difference was observed in unadjusted Kaplan-Meier plots of kidney transplant rejection/failure for black kidney transplant patients in England versus New York State (see figure 1). After adjustment for baseline variables, Cox regression analysis confirmed black kidney

**Table 3** Adjusted logistic regression analysis of kidney transplant mortality in both England and New York State by ethnicity between 2003 and 2014*

| Ethnicity | England | | New York State | |
| --- | --- | --- | --- | --- |
| | OR (95% CI) | p Value* | OR (95% CI) | p Value* |
| **30-Day mortality** | | | | |
| Black | 1 (baseline group) | | 1 (baseline group) | |
| Other | 1.48 (0.76 to 2.88) | 0.246 | 0.79 (0.55 to 1.15) | 0.214 |
| White | 0.81 (0.44 to 1.49) | 0.491 | 0.62 (0.44 to 0.86) | 0.005 |
| **1-Year mortality** | | | | |
| Black | 1 (baseline group) | | 1 (baseline group) | |
| Other | 1.27 (0.85 to 1.90) | 0.234 | 0.79 (0.63 to 0.99) | 0.064 |
| White | 0.94 (0.65 **to** 1.34) | 0.728 | 0.66 (0.55 to 0.81) | <0.001 |

*Adjusted for age, sex, admission method and number of emergency readmissions.

transplant patients in New York State versus England had over the double the HR for kidney transplant rejection/failure during follow-up (HR 2.15, 95% CI 1.91 to 2.43) (see table 4), which was identical to a similar analysis focusing only on white patients (see online supplementary table 1).

## DISCUSSION

Our comparative analysis of two large population-based cohort data sets demonstrates that black kidney transplant patients have different baseline characteristics and post-transplant outcomes in England compared with New York State. Most importantly, black patients in England have lower mortality than their New York State counterparts, where black ethnicity in the latter cohort was found to be independently associated with increased 30-day and 1-year mortality. Our analysis, to our knowledge, is the first comparative study comparing black kidney allograft

outcomes in contemporaneous population cohorts. Our study highlights significant differences between black kidney transplant patients in England versus New York State and suggests caution in translating outcomes for black transplant patients between different countries.

The strengths and weaknesses of this study must be appreciated for the correct interpretation of our results. There are likely to be numerous confounders that have an impact on black mortality post kidney transplantation that we were unable to factor in (eg, smoking, lifestyle, sociocultural factors and dialysis vintage). It should be acknowledged that New York State data may not be completely representative of other states in the USA, and we are crudely inferring that data from New York State is broadly representative of the USA. Missing data (and misclassification bias) also have an implication on the analyses performed, and this limitation is inherent with all epidemiological studies of this

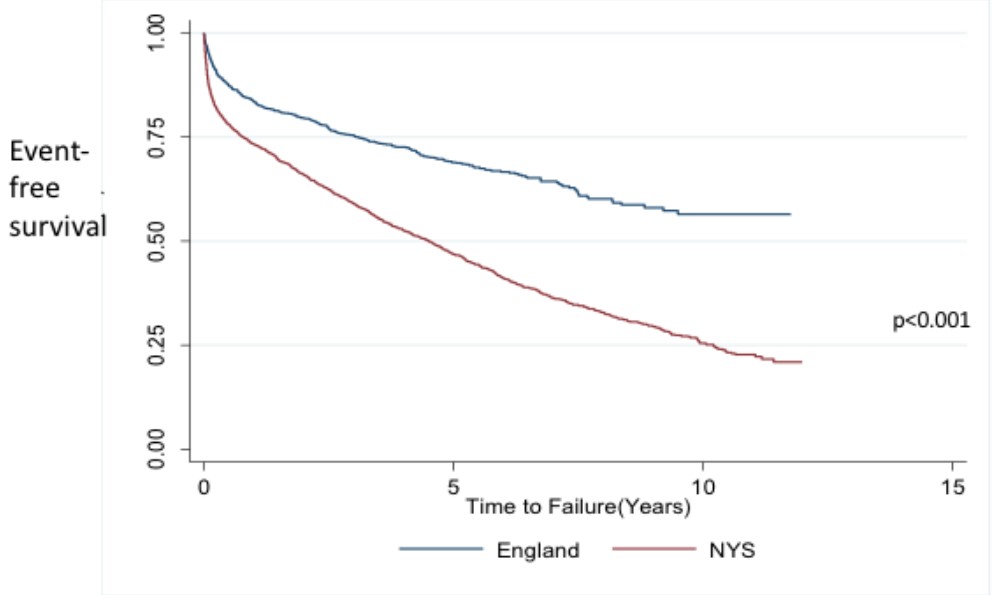

**Figure 1** Unadjusted Kaplan-Meier curve of kidney transplant rejection/failure for black kidney transplant patients in England versus New York State (NYS) between 2003 and 2014 (log-rank test, p <0.001).

**Table 4** Adjusted Cox regression analysis for transplant rejection/failure among black kidney transplant patients in England and NYS between 2003 and 2014

| | HR (95% CI) | p Value |
|---|---|---|
| Age | 1.00 (0.99 to 1.00) | 0.171 |
| Sex | | |
| Male | 1 (baseline group) | |
| Female | 1.00 (0.91 to 1.09) | 0.928 |
| Type of donor | | |
| Alive | 1 (baseline group) | |
| Dead | 1.37 (1.22 to 1.54) | <0.001 |
| Unknown | 1.08 (0.92 to 1.27) | 0.339 |
| Diabetes | 1.09 (0.98 to 1.20) | 0.114 |
| Acute MI | 0.99 (0.84 to 1.17) | 0.935 |
| CVD | 1.04 (0.87 to 1.24) | 0.697 |
| PVD | 1.16 (0.98 to 1.37) | 0.086 |
| CHF | 1.16 (1.04 to 1.30) | 0.008 |
| Year | | |
| Pre-2007 | 1 (baseline group) | |
| Post-2007 | 1.30 (1.16 to 1.45) | <0.001 |
| Country | | |
| England | 1 (baseline group) | |
| NYS (USA) | 2.15 (1.91 to 2.43) | <0.001 |

CHF, congestive heart failure; CVD, cerebrovascular disease; MI, myocardial infarction; NYS, New York State.

type. Unfortunately, while the majority of data obtainable was matched between the two cohorts, we were unable to extract any information regarding socioeconomic deprivation status or cause of death from the SPARCS data set. Our utilisation of the composite for kidney transplant rejection/failure from administration records is an inferior surrogate compared with registry-derived data regarding kidney allograft rejection and graft failure. The ability to extract more data, and to link data sets to create more comprehensive databases, allows the minimisation of confounding and provides a more robust platform from which to conduct meaningful epidemiological analysis. Further work in this area is clearly warranted and should be undertaken using more robust renal data including registry data from both the USA (eg, USRDS and Scientific Registry of Transplant Recipients) and the UK (eg, UK Transplant Registry and Renal Registry). Finally, mortality data are readily available and were the sole analytical endpoint of this analysis. However, absence of recorded death may not necessarily translate to assumption that a recipient remains alive (eg, lost to follow-up due to emigration).

Current literature suggests that black kidney transplant patients in the USA have inferior kidney allograft survival compared with white recipients,[4] and risk for death is magnified among black American versus Canadian kidney transplant patients.[7] Reassuringly, recently published work demonstrates significant improvement in post kidney transplant outcomes in the USA over the last two decades, more so for black versus white patients, leading to reduced disparity.[5] This may be related to advances in immunosuppression and post-transplant management, possibly benefiting black kidney transplant patients more significantly who perhaps were more disproportionately overburdened by immunological barriers in the past. While some disparity remained in their analysis in relation to significantly raised adjusted HRs for black patients and risk for graft loss in the contemporary setting, the vastly improved outcomes were encouraging for transplant clinicians to actively promote kidney transplantation among black patients. Our analysis suggested worse kidney allograft outcomes since 2007 for all black kidney transplant recipients, which may be due to more high-risk kidney transplantation of medically complex or immunological difficult patients, but this requires further analysis with more transplant-specific data as it may simply reflect better administrative coding of kidney rejection/failure since 2007.

By contrast, no mortality difference between black and white patients has been observed in the few limited publications within European transplant centres.[8–10] Our data therefore support the inference from existing literature about disparate outcomes for black patients in the USA versus elsewhere and raises two important questions: (1) Why outcomes for black transplant patients in the USA remain inferior to white patients, and (2) Why black transplant patients in the USA have inferior outcomes in comparison with black patients outside of the USA. It could be argued that the Black population in England differs from the USA, but this is an insufficient explanation as black people in Canada and the USA share geographical origins but still have dissimilar mortality outcomes.[7] It is important to distinguish the significant differences in baseline demographics between our two black cohorts. For example, medical comorbidities such as history of diabetes and atherosclerotic diseases were less common among black transplant recipients in England compared with New York State, suggesting a higher level of cardiovascular burden at baseline for the latter. This could explain the significantly increased risk for cardiac events within the first year after kidney transplantation for black kidney transplant patients in New York State versus England. Due to the absence of cause of death from the SPARCS data, we were unable to ascertain the nature of deaths among black patients in New York State, but an increase in cardiac deaths would be a reasonable assumption. However, baseline demographics alone cannot account for the significant difference in mortality.

Perhaps the most fundamental difference between countries is the infrastructure delivering and providing healthcare. Black kidney transplant patients in England have universal health coverage as part of the taxpayer-funded NHS. This ensures complete financial access to immunosuppressive medications and other aspects of care including allied medications (eg, antihypertensive agents, glucose-lowering agents and lipid-lowering

drugs) and long-term medical follow-up to support post-transplant care. Compared with the USA, countries such as England, Canada and France that demonstrate similar mortality outcomes after transplantation for black versus white patients share elements of universal health coverage,[6 8–10] and this is an obvious factor to explain disparate mortality outcomes. This is also supported by our analysis also showing improved graft survival for white patients in England versus New York States, suggesting our analysis reflects better outcomes overall in England rather than any ethnicity effect. However, universal health coverage alone is unlikely to be the sole explanation for differing outcomes. Prospective clinical trials in the USA, which may attenuate the risk of any lack of access, still demonstrate poorer outcomes for black patients in subgroup analyses,[15] suggesting health coverage alone does not fully explain poor outcomes. This is supported by evidence within the USA shown by Chakkera et al in their exploration of racial disparities using kidney transplant cohorts, which included patients receiving healthcare within versus outside the Department of Veterans Affairs (VA).[16] The VA is the largest integrated managed health-care system in the USA and provides comprehensive medical care to eligible veterans including coverage of prescription drugs. Perhaps surprisingly, the association of black ethnicity with poorer outcomes was consistent in non-VA versus VA users, with the latter associated with increased risk of graft loss (adjusted graft failure 1.31, 95% CI 1.10 to 1.57) and death (adjusted mortality 1.10, 95% CI 0.90 to 1.34). While noting adjusted mortality for VA users was not significantly raised in the analysis (likely due to a smaller cohort of 1646 VA users versus 77 715 non-VA users), the adjusted risks for graft failure and death were almost identical between VA and non-VA users (1.31 and 1.31, respectively, for graft failure and 1.10 and 1.11, respectively, for mortality).

This raises the question as to what underlies poor outcomes among black kidney transplant patients in the USA while acknowledging disparate outcomes appears to be narrowing between black and white patients' post kidney transplantation.[5] Factors such as non-adherence, which could be a surrogate for non-affordability of drugs in the USA, is a phenomenon that is common in transplantation cohorts everywhere (including countries with universal health coverage).[17] There could be selection bias from poor access to transplantation for black people, which has been acknowledged in transplant centres across the world and has been comprehensively reviewed elsewhere.[18] However, from the most up-to-date census reports, individuals of black ethnicity make up 3.5% of the English population[19] and 17.6% of the New York State population[20] and black patients formed 6.5% and 23.0% of our transplant cohorts, respectively. While this suggests black patients are over-represented as kidney allograft recipients in both England and New York State, respectively, it must be remembered that ESKD is more prevalent in black communities, and therefore they are more likely to need renal replacement therapy. In should

be noted that disparate post-transplant outcomes between Europe and the USA are not unique to kidney allograft recipients of black ethnicity alone. Gondos et al, analysing data from the United Network for Organ Sharing and the Collaborative Transplant Study, compared kidney transplant outcomes in the USA versus Europe, respectively.[21] Adjusted HRs for graft loss were higher in the USA compared with Europe and identifying the contributing factors to this disparity should be considered a high priority to improve the delivery of post-transplant care. Socioeconomic status, education and poverty are all likely to be important, and black transplant patients in the USA may be an inherent risk for higher mortality or have weaker social support networks or negative health behaviour that drives risk for adverse outcomes (such as smoking, alcohol consumption, sedentary lifestyle and poor diet).[22–24] It is clear that our understanding of the social epidemiological impact on health inequalities is limited,[25] but deepening our understanding of these issues may further reduce the disparity in mortality outcomes seen in the USA for black kidney allograft recipients.

Our analysis should be interpreted in the context of the well-recognised observation of reduced risk for death for black versus white ESKD patients after commencement of dialysis, and this phenomenon has been noted in both the USA[2] and the UK.[3] However, inferring that all black patients on dialysis have reduced risk for death on dialysis may be incorrect when interpreting large population cohorts in the context of competing factors. For example, Kucirka et al conducted an observational cohort study of 1 330 007 incident end-stage renal disease patients in the USRDS and showed black patients have lower mortality on dialysis than white patients (adjusted HR 0.84; 95% CI 0.83 to 0.84; p<0.001).[26] However, after stratification by age and treating kidney transplantation as a competing risk, black patients below the age of 50 had significantly higher mortality than their white counterparts on dialysis. Teasing out the differences that help explain our disparate mortality outcomes between black patients in England versus New York State will likely prove more challenging than just a simple explanation. Caution must also be exercised in the interpretation of epidemiological data to ensure false inferences are not made from statistical analyses. For example, Yeates et al found black kidney transplant patients in Canada had significantly lower post-transplantation mortality compared with white patients (HR 0.49; 95% CI 0.28 to 0.88; p=0.02).[6] This finding is surprising, and the only documented example of this observation, which requires corroboration before any legitimate interpretation, can be made. However, there are recognised steps that can be taken to reduce the risk of post-transplant mortality for black patients including greater use of living-kidney donors, which has been shown to improve outcomes and to reduce racial disparity of post-transplant outcomes.[27]

To conclude, this is the first comparative study examining mortality for black kidney transplant patients

between two different countries and found black kidney transplant patients in New York State have a greater hazard for death and kidney transplant failure/rejection compared with their black counterparts in England. Our results suggest that further work is essential to investigate poor outcomes for black patients after kidney transplantation in the USA to aid our inadequate understanding of their social, environmental and/or biological pressures. While macro-level factors like universal health coverage may be an important factor contributing to post-transplantation mortality and allograft loss for black patients, our study suggests additional micro-level factors are likely to be concomitantly important and are in need of deeper evaluation.

**Contributors** ST, JN and AS designed the study; ST, HG, FJS, JM and FE contributed on data extraction; ST, JM and FE contributed on data analysis; ST, JN and AS interpreted the data; ST and AS wrote the original draft; and ST, HG, FJS, JN, JM, FE and AS reviewed the manuscript.

**Funding** We are grateful for the funding in support of the BMedSci intercalated degrees for ST and FJS (from Kidney Research UK) and HG (from Arthur Thomson Trust).

**Competing interests** None declared.

**Ethics approval** Clinical audit office.

**Provenance and peer review** Not commissioned; externally peer reviewed.

**Data sharing statement** No additional data available.

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
