## [Reviewer comments · BMJ Open]

ARTICLE DETAILS

TITLE (PROVISIONAL)	Do outcomes after kidney transplantation differ for Black patients in England versus New York State? A comparative, population-cohort analysis
AUTHORS	Tahir, Sanna; Gillott, Holly; Jackson-Spence, Francesca; Nath, Jay; Mytton, Jemma; Evison, Felicity; Sharif, Adnan

VERSION 1 - REVIEW

REVIEWER	Jennifer Gander Emory University School of Medicine
REVIEW RETURNED	25-Oct-2016

GENERAL COMMENTS	African American (also known as black) transplant recipients have been previously shown to have increased risk for negative post-transplant outcomes compared to white transplant recipients. The authors investigated whether black kidney transplant recipients in England have disparate outcomes versus black kidney transplant recipients in New York State. The authors present a comparative cohort study that examines 30-day mortality, 1-year mortality, and graft failure in black transplant recipients in New York state and England. Among the New York State cohort, authors report white transplant recipients with a lower 30-day mortality and 1-year mortality compared to black transplant recipients. There was no significant association between race and mortality outcomes among the England cohort. Overall the study was well designed and executed. The authors provided sufficient detail throughout the article. Major concerns include outcome definition and statistical analysis. The outcomes, especially graft failure, need to be clarified within the methods. Details regarding logistic regression and survival analysis are unclear. It is unclear which analysis was used to investigate the associations with the primary outcomes. These concerns as well as minor concerns are detailed within the attachment. I suggest the article be accepted with minor revisions.
---

REVIEWER	Babak Orandi University of California, San Francisco U.S.A.
REVIEW RETURNED	30-Oct-2016

GENERAL COMMENTS	This is a well-done paper that answers an important question about differences in outcomes for Black patients in the US vs. England and generates many new questions about the etiology of these differences. The paper includes a very honest and transparent appraisal of the study's limitations and a very interesting and robust
--

discussion of the authors' findings.

A few points to address:

The authors excluded patients from the analysis if they were missing demographic data and assume that they were missing at random, which may not be a safe assumption. I think it is important to know how many patients from each country were excluded because they were missing race data 1) because it is the main variable of interest and 2) because it frequently is missing from large data sets. If many patients were excluded because of missing data on race, then it may be necessary to do some sensitivity analyses (for example, assume all patients with missing race are Black and then assume they are all white to see if it alters the inferences of the paper).

In Table 2, it says that 20.6% of NYS Black patients had an unknown donor type (deceased vs. living)? That seems like a very high number and could potentially introduce significant bias into the results. Perhaps a sensitivity analysis assuming that the unknown donor types are all deceased and then are all living donors might provide some assurance about the results that are obtained.

While it doesn't necessarily need to be shown in the manuscript, I think it would be important to repeat the analyses comparing only deceased donors and only living donors. There may be some confounding by indication here. The authors point out that Black patients in England are more likely to get living donors. Because living donors are done electively, the donors AND the recipients have had recent work ups that allow them to be optimized and in some cases, the recipients are deemed non-transplant candidates because of comorbidities or deteriorations in their health. This up-to-date work-up is a luxury that is frequently not possible for recipients of deceased donor kidneys. This may not fully be captured by including a binary variable of deceased vs living donor in the regression model and stratifying the analyses allows us to compare apples to apples and oranges to oranges.

Is length of time on dialysis not available?

The sentence "Black transplant patients in the United States have an inherent risk for higher mortality or have weaker social support networks or negative health behavior that drives risk for adverse outcomes (such as smoking, alcohol consumption, sedentary lifestyle and poor diet)." should have some references, particularly vis a vis weaker social support networks.

For figure 1, is it possible to separate out graft failure and rejection, rather than treat them as a composite endpoint because the gravity of developing rejection is much lower than graft failure?

Table 4—in the Cox regression analysis, was there a reason that age was treated as a binary variable (<50 vs 50+) rather than a continuous one? If there are major differences in the transplant practice patterns between the two countries based on age (and potentially by race as well), those subtleties are lost by making age a binary variable.

REVIEWER	Natalie Staplin Senior Statistician Nuffield Department of Population Health, University of Oxford United Kingdom
REVIEW RETURNED	21-Nov-2016

GENERAL COMMENTS	The choice of Cox proportional hazards model for time to event variables and logistic regression for categorical variables is appropriate, however the method used to examine the relevance of black ethnicity to survival by geographical region is flawed. This has the potential to change the interpretation of the results so is an important point to address. See below for detailed comments on some proposed alternative analyses and a few other issues with the statistical methods. MAJOR COMMENTS  1. Currently in the manuscript, it is stated that the relevance of black ethnicity to mortality differs between England and New York State because the hazard ratio for white ethnicity compared to black ethnicity was significantly lower in the model for New York State but not significant in the model for England. The correct way to test whether the hazard ratio for white vs black ethnicity differs by geographical region is to include an interaction term between ethnicity and geographical region in the model and test whether this is significant using a likelihood ratio test. 2. Could you clarify whether the analyses in Table 2 have been adjusted for any covariates? If they have not, then this is probably not the best approach to compare mortality outcomes for black patients in New York State and England. Ideally, regression models should be used for these comparisons to adjust for differences in important covariates such as age and co-morbidities. 3. It is good to see that you have considered whether the proportional hazards assumption is valid. However, given the number of variables included in the model (which includes a time updated covariate); this assumption should be checked by testing whether the scaled Schoenfeld residuals have a non-zero slope. 4. In the second paragraph on page 9, it is stated that 'Any missing data was assumed to be missing at random and subjected to simple listwise deletion from the analysis'. This method of handling missing data is only unbiased if the data is missing completely at random, which is a very strong assumption. Also, depending on the amount of missing data, this approach can result on a considerable loss of power. How many participants were excluded due to missing data? You may want to consider imputing values for these individuals instead. 5. Recording of ethnicity in Hospital Episode Statistics is not always complete (particularly earlier in the cohort) and if those with missing values are excluded from the analyses then it is possible that the sample used in these analyses is not representative of the kidney transplant population in England. This should be acknowledged in the discussion. 6. At the bottom of page 8, it is stated that Mann-Whitney tests were used for all continuous variables to compare differences between groups. This test is most appropriate for variables that are non-normally distributed or have extreme outliers. If variables were normally distributed, then analysis of variance would be a more powerful test of differences between groups. 7. ICD10 and OPCS-4 codes should be provided for all the secondary outcomes. MINOR COMMENTS
--

	8. In Table 4, age is included as a binary variable (<50 vs 50+) which is a crude method of adjusting for age. Have you considered including age as a continuous variable? 9. Why was 2007 chosen as the cutpoint for year of transplant? Is it possible to adjust more finely for year of transplant by including more categories in this variable? 10. The title of Figure 1 suggests that the curves are estimated using Cox regression but on page 12 it states that they are unadjusted Kaplan-Meier plots. 11. At the end of the first paragraph on page 9, the phrase 'time variate covariate' is used, do you mean 'time updated covariate' here?
--	--

REVIEWER	John Kalbfleisch University of Michigan United States
REVIEW RETURNED	22-Nov-2016

GENERAL COMMENTS	This is an interesting paper that compares the post-transplant survival outcomes of Black patients in the UK with those of Black patients treated in New York State. These results are surprising and raise a number of questions as to why these geographical differences should occur. The analyses appear to be reasonably well done and there is some good discussion. The paper, however, also contains a great deal of speculation. Some comments:  1. It would seem more sensible also to compare White patients in the NY and England. I suspect that there are no large differences between the outcomes for White patients, but that does not seem to be clearly covered in the paper. 2. I wondered about the choice of the SPARCS data instead of the SRTR / UNOS data, which would also provide follow up on a national rather than a state wide basis. Can you comment on the comparative accuracy of these data bases and why the choice of SPARCS was made. 3. On Page 9, lines 8 and 9, it is stated the "Recipients identified as having kidney transplant rejection/failure were added as a time variate covariate in the logistic model for mortality." I am not sure what this statement means, but if it is stating that graft failure is included in mortality models as a covariate, then this is not a good thing to do. Graft failure is highly predictive of subsequent mortality and if one population has a higher rate of graft failure, adjusting for that could remove or create a mortality difference. This needs explanation and may require reanalysis. 4. In the Introduction and elsewhere it is remarked that it is unclear as to whether outcomes of Black patients in one country generalize to another, although this seems clear on general grounds, especially with the difference between black and white patients in the US, which is not seen elsewhere. It is also remarked that this is the first comparison between countries of post-transplant survival of black patients, yet the Canadian/US study is also referenced, which would seem to contradict the claim. 5. Covariates collected and used in the analyses do not include some common covariates like BMI or vintage. In addition, there seem to be no covariates on the donor except for living versus deceased. With donors, there are other aspects such as age, cause of death, race, comorbidities etc. that are important in predicting post-transplant outcomes. Were such data unavailable? Is there
--

	potential for such covariates to explain at least some of the differences seen. Also, when were the comorbidities determined? Are these comorbidities at baseline (time of transplant) or determined over time prior to transplantation? Some explanation is needed. 6. The first sentence on page 10 needs revision. This material should refer to table 1. A comparison of demographics for White patients would also be interesting. 7. Post-transplant outcomes. Are the differences in the relative risks for Whites reported in Table 2 for NY versus England significantly different? Having White as the baseline in Table 2 would seem better to me. This would also lessen the apparent differences in relative risks for Other races. 8. Top of page 13. Why make this crude inference. Why not just compare NY with England? You could have used SRTR data for a broader comparison. 9. I found the discussion section to be particularly disjointed and unfocused. I think it would be greatly improved with a careful rewriting that reduced its length by about half and avoided repetition of points already made (often several times). A presentation with shorter more focused paragraphs would be much better. Some of the discussion is very speculative – for example the material at the bottom of page 13 and top of page 14 beginning with “This may be related to advances...”. I also found the last paragraph on page 15 confusing. Maybe a section on Study Limitations would help to organize the discussion. 10. On page 17, I wonder about the statement “However, after stratification by age and treating kidney transplantation as a competing risk,.....” I’d guess that this analysis is of cumulative incidence and may simply be a result of a higher rate of transplantation among black patients in England. It would be better to look at the rate of failure among patients alive with functioning graft (e.g. a cause specific hazards analysis).
--	---

VERSION 1 – AUTHOR RESPONSE

Reviewer: 1

Reviewer Name: Jennifer Gander

Institution and Country: Emory University School of Medicine Please state any competing interests or state 'None declared': None declared

Please leave your comments for the authors below:

African American (also known as black) transplant recipients have been previously shown to have increased risk for negative post-transplant outcomes compared to white transplant recipients. The authors investigated whether black kidney transplant recipients in England have disparate outcomes versus black kidney transplant recipients in New York State.

The authors present a comparative cohort study that examines 30-day mortality, 1-year mortality, and graft failure in black transplant recipients in New York state and England. Among the New York State cohort, authors report white transplant recipients with a lower 30-day mortality and 1-year mortality compared to black transplant recipients. There was no significant association between race and mortality outcomes among the England cohort.

Overall the study was well designed and executed. The authors provided sufficient detail throughout the article. Major concerns include outcome definition and statistical analysis. The outcomes, especially graft failure, need to be clarified within the methods. Details regarding logistic regression

and survival analysis are unclear. It is unclear which analysis was used to investigate the associations with the primary outcomes. These concerns as well as minor concerns are detailed within the attachment. I suggest the article be accepted with minor revisions.

Thank you for these comments. The Methods section has now been extensively updated and edited to reflect these concerns.

JOURNAL: BMJ Open

TITLE: Do outcomes after kidney transplantation differ for Black in England versus New York State? A comparative, population-cohort analysis

FIRST AUTHOR: Tahir

OVERVIEW:

African American (also known as black) transplant recipients have been previously shown to have increased risk for negative post-transplant outcomes compared to white transplant recipients. The authors investigated whether black kidney transplant recipients in England have disparate outcomes versus black kidney transplant recipients in New York State.

The authors present a comparative cohort study that examines 30-day mortality, 1-year mortality, and graft failure in black transplant recipients in New York state and England. Among the New York State cohort, authors report white transplant recipients with a lower 30-day mortality and 1-year mortality compared to black transplant recipients. There was no significant association between race and mortality outcomes among the England cohort. Overall the study was well designed and executed.

The authors provided sufficient detail throughout the article. Major concerns include outcome definition and statistical analysis. The outcomes, especially graft failure, need to be clarified within the methods. Details regarding logistic regression and survival analysis are unclear. It is unclear which analysis was used to investigate the associations with the primary outcomes. These concerns as well as minor concerns are detailed below. I suggest the article be accepted with minor revisions.

Thank you for these comments. The Methods section has now been extensively updated and edited to reflect these concerns.

GENERAL COMMENTS

p-values with 95% CI - p-values are not necessary when 95% Confidence Intervals can be presented.

These have now been removed.

ABSTRACT

Settings and Participants- Were the participants 1st kidney transplants?

Yes – this has now been added to the text.

Statistical Analysis- Please provide 1 sentence that describes the statistical analysis used to investigate the relationship between race, place of origin, and outcome.

The methods section has now been altered.

Results: Inpatient mortality - Avoid using 'equivalent' unless the mortality was equal. Substitute 'equivalent' for ~'similar' or 'not statistically different'

This has now been changed accordingly.

Results: "...transplant rejection/failure was lower among English versus Black patients in New York State." - Please clarify. Did you mean that black transplant recipients in England vs black recipients in New York State?

Yes – this has now been amended accordingly.

INTRODUCTION

P5 USRDS 2010 incident rates- More updated USRDS data is available. Currently the 2013 incidence rates are available in the 2015 Annual Report.

Thank you for this comment. We have not utilised SRTR or USRDS data however we would encourage further work in this area by researchers using these more robust renal transplant data sets.

METHODS

P6 Study Population - Please clarify in the first sentence these are adult kidney- alone transplants. First kidney transplant or were participants with a 2nd, 3rd, etc transplant included?

This has now been done.

P7 Death data- Nice information on the linkage to obtain death data. Please clarify the dates of the death data.

Thank you – this has now been done.

P8 Data inclusion- Please provide the number of initial patients considered, and then the number of patients excluded for each reason.

This has now been done.

P8 Outcomes- Was this all-cause mortality or cause specific? Please clarify. How was 'risk of re-hospitalization' defined and captured? Were all secondary outcomes obtained through the medical record? How was 'transplant rejection/failure' defined? Was this rejection with 30- days of transplant?

We have now edited our Methodology section to provide all this information for clarification.

P8-9 Statistical Analysis: Cox model Provide more detail on when survival analysis was used, beyond the stated "where time to event data was available." Which events/outcomes were assessed using Cox? Were all patients included in these analyses? Which associations were investigated with Cox and which were investigated with logistic regression? As it is currently written, it leads the reader to think that Cox was only used for some patients, while logistic regression was used for other patients. Thank you for providing a comment on the proportional hazard assumptions.

This has now been added to the Methods section.

P9 Statistical Analysis: Covariables Were you able to control for time on dialysis? Time on dialysis is a strong factor that predicts poor post-transplant outcomes. Previous research has shown the disparity (African American vs white) in post-transplant outcomes may be associated with African American's longer time on dialysis prior to transplantation.

We were not able to control for time on dialysis as this information was not available to us from the

existing data sets.

P9 Statistical Analysis: time to graft failure Authors state the time to graft failure was treated as a covariate in mortality analysis. The time to graft failure variable needs to be defined previous to the statistical analysis section.

Thank you for this comment – this has now been done.

P9 Missing data What is the percent of missing data?

We have now added data on number of exclusions.

P8 Ethical review Please clarify what is meant by 'pseudoanonymized nature' regarding ethical approval. Was the data de-identified?

Yes - this was de-identified data with no patient identifiable information.

RESULTS

P10 Baseline demographics: Please provide the % (or mean) and pvalue within each comparison. For example: "...recipients in England versus New York state were found to be younger (46.3 vs 50.1, p-value<0.001)...."

This has now been done.

Post-transplant outcomes: Please provide the % (or mean) and pvalue within each comparison. Similar to described above.

This has now been done.

Post-transplant outcomes: Present the mortality outcomes prior to presenting the secondary outcomes such as cardiac events.

This has now been edited accordingly.

P11 Regression Analysis: Logistic model for 30-day and 1-year mortality: Was this an adjusted model? If so, what variables were included in the final model.

Although not statistically significant, please provide to OR and 95% CI for 30-day and 1-year mortality for English African American transplant recipients.

These were adjusted models and we have now provided OR and 95% CI for 30-day and 1-year mortality data.

P12 Cox regression comparing graft failure Authors need to define this variable earlier in Methods. Was this graft failure in 30-days? 1-year? Over total followup? Was this adjusted analysis?

This has now been done.

P12 Cox regression: Table 4: Within text states that graft failure was compared between black transplant recipients in England vs New York state. The text does not seem to match up from Table 4.

We have checked through the Tables and text to ensure everything matches

DISCUSSION

P12-13 Limitations Nice presentation of strengths and limitations. Please provide potential future directions to the limitations presented.

This has been added.

Limitations of Data The authors could present that a future direction be to obtain and use the United States Renal Data System that is more representative of US end state kidney disease patients than data just out of New York state.

We completely agree with this statement and have added this to our discussion.

Limitations One limitation is the lack of time on dialysis. Please state the limitation, potential implications, and future directions.

Thank you and we agree with this limitation – we have added this to our discussion.

TABLES/FIGURES

General Footnotes: Please provide footnotes to describe abbreviations (ex: NYS), describe which statistical test(s) were used to obtain p-values, covariables included in adjusted logistic regression (Table 3)

This has been added.

General Titles Please provide the years of data included in each table/figure

This has been added

Table 3 Format Authors should consider revising the format of Table 3 to put England and New York State in side-by-side columns.

This has been done.

Table 4 The outcome and comparison is not clear. Is the analysis an adjusted/bivariable hazard model among the entire cohort (England + New York state)?

We have amended the Title to make this clearer.

Figure 1 Kaplan Meier Curves Authors stated within the text there was a significant difference between the England vs New York State KM curves. Please provide the p-value of comparison on the figure.

This has been added.

Reviewer: 2

Reviewer Name: Babak Orandi

Institution and Country: University of California, San Francisco, U.S.A.

Please state any competing interests or state 'None declared': None declared

Please leave your comments for the authors below:

This is a well-done paper that answers an important question about differences in outcomes for Black patients in the US vs. England and generates many new questions about the etiology of these differences. The paper includes a very honest and transparent appraisal of the study's limitations and a very interesting and robust discussion of the authors' findings.

A few points to address:

The authors excluded patients from the analysis if they were missing demographic data and assume that they were missing at random, which may not be a safe assumption. I think it is important to know how many patients from each country were excluded because they were missing race data 1) because it is the main variable of interest and 2) because it frequently is missing from large data sets. If many patients were excluded because of missing data on race, then it may be necessary to do some sensitivity analyses (for example, assume all patients with missing race are Black and then assume they are all white to see if it alters the inferences of the paper).

Thank you for this comment. We have now undertaken some additional sensitivity analyses and they are attached as an additional file for review.

In Table 2, it says that 20.6% of NYS Black patients had an unknown donor type (deceased vs. living)? That seems like a very high number and could potentially introduce significant bias into the results. Perhaps a sensitivity analysis assuming that the unknown donor types are all deceased and then are all living donors might provide some assurance about the results that are obtained.

Thank you for this comment. We have now undertaken some additional sensitivity analyses and they are attached as an additional file for review.

While it doesn't necessarily need to be shown in the manuscript, I think it would be important to repeat the analyses comparing only deceased donors and only living donors. There may be some confounding by indication here. The authors point out that Black patients in England are more likely to get living donors. Because living donors are done electively, the donors AND the recipients have had recent work ups that allow them to be optimized and in some cases, the recipients are deemed non-transplant candidates because of comorbidities or deteriorations in their health. This up-to-date work-up is a luxury that is frequently not possible for recipients of deceased donor kidneys. This may not fully be captured by including a binary variable of deceased vs living donor in the regression model and stratifying the analyses allows us to compare apples to apples and oranges to oranges.

Thank you for this comment. We have now undertaken some additional sensitivity analyses and they are attached as an additional file for review.

Is length of time on dialysis not available?

Unfortunately, this data is not available from the administrative data sources and is cited as a limitation of this analysis in the discussion (2nd paragraph).

The sentence "Black transplant patients in the United States have an inherent risk for higher mortality or have weaker social support networks or negative health behavior that drives risk for adverse outcomes (such as smoking, alcohol consumption, sedentary lifestyle and poor diet)." should have some references, particularly vis a vis weaker social support networks.

Thank you for this comment. We have edited this section and also amended the statement by

suggesting Black transplant patients' may have these inherent risks as we believe definitive and robust data on these sociological issues are awaited.

For figure 1, is it possible to separate out graft failure and rejection, rather than treat them as a composite endpoint because the gravity of developing rejection is much lower than graft failure?

Unfortunately, we did not have access to transplant registry data to extract hard data on rejection and graft failure. We explored an administrative code for kidney transplant rejection/failure to classify this combined outcome, which cannot be separated out individually into allograft rejection and failure.

Table 4—in the Cox regression analysis, was there a reason that age was treated as a binary variable (<50 vs 50+) rather than a continuous one? If there are major differences in the transplant practice patterns between the two countries based on age (and potentially by race as well), those subtleties are lost by making age a binary variable.

Thank you for this comment – we have amended our data accordingly.

Reviewer: 3

Reviewer Name: Natalie Staplin

Institution and Country: Senior Statistician, Nuffield Department of Population Health, University of Oxford, United Kingdom Please state any competing interests or state 'None declared': None declared

Please leave your comments for the authors below:

The choice of Cox proportional hazards model for time to event variables and logistic regression for categorical variables is appropriate, however the method used to examine the relevance of black ethnicity to survival by geographical region is flawed. This has the potential to change the interpretation of the results so is an important point to address. See below for detailed comments on some proposed alternative analyses and a few other issues with the statistical methods.

MAJOR COMMENTS

1. Currently in the manuscript, it is stated that the relevance of black ethnicity to mortality differs between England and New York State because the hazard ratio for white ethnicity compared to black ethnicity was significantly lower in the model for New York State but not significant in the model for England. The correct way to test whether the hazard ratio for white vs black ethnicity differs by geographical region is to include an interaction term between ethnicity and geographical region in the model and test whether this is significant using a likelihood ratio test.

Thank you for this comment. We have now made it clearer in our methodology what we have undertaken. We have not undertaken the interaction term as our model did not include white and black ethnicity together in the analysis and only black ethnicity was compared between geographical areas. Therefore, interaction analysis was not required.

2. Could you clarify whether the analyses in Table 2 have been adjusted for any covariates? If they have not, then this is probably not the best approach to compare mortality outcomes for black patients in New York State and England. Ideally, regression models should be used for these comparisons to adjust for differences in important covariates such as age and co-morbidities.

We have now updated our methods accordingly to make this clearer.

3. It is good to see that you have considered whether the proportional hazards assumption is valid. However, given the number of variables included in the model (which includes a time updated

covariate); this assumption should be checked by testing whether the scaled Schoenfeld residuals have a non-zero slope.

Thank you for this comment. We have now updated our methods and re-done our analysis accordingly.

4. In the second paragraph on page 9, it is stated that 'Any missing data was assumed to be missing at random and subjected to simple listwise deletion from the analysis'. This method of handling missing data is only unbiased if the data is missing completely at random, which is a very strong assumption. Also, depending on the amount of missing data, this approach can result on a considerable loss of power. How many participants were excluded due to missing data? You may want to consider imputing values for these individuals instead.

Thank you for this comment. We have now updated our methods and re-done our analysis accordingly.

5. Recording of ethnicity in Hospital Episode Statistics is not always complete (particularly earlier in the cohort) and if those with missing values are excluded from the analyses then it is possible that the sample used in these analyses is not representative of the kidney transplant population in England. This should be acknowledged in the discussion.

These missing values were not excluded.

6. At the bottom of page 8, it is stated that Mann-Whitney tests were used for all continuous variables to compare differences between groups. This test is most appropriate for variables that are non-normally distributed or have extreme outliers. If variables were normally distributed, then analysis of variance would be a more powerful test of differences between groups.

Thank you for this comment. We have now updated our methods and re-done our analysis accordingly.

7. ICD10 and OPCS-4 codes should be provided for all the secondary outcomes.

These have now been added.

MINOR COMMENTS

8. In Table 4, age is included as a binary variable (<50 vs 50+) which is a crude method of adjusting for age. Have you considered including age as a continuous variable?

This has now been done.

9. Why was 2007 chosen as the cutpoint for year of transplant? Is it possible to adjust more finely for year of transplant by including more categories in this variable?

2007 was chosen as an appropriate cutpoint year due to being a landmark for major change in immunosuppression with publication of the SYMPHONY trial (Ekberg et al. NEJM 2007) when many transplant units switched their primary immunosuppressant from ciclosporin to tacrolimus. For clarity of display, we opted not to sub-divide further for more finer analysis which would reduce power of the analysis and be subjective in nature.

10. The title of Figure 1 suggests that the curves are estimated using Cox regression but on page 12

it states that they are unadjusted Kaplan-Meier plots.

We have now updated this.

11. At the end of the first paragraph on page 9, the phrase 'time variate covariate' is used, do you mean 'time updated covariate' here?

This has been deleted – we did not undertake time variate analysis.

Reviewer: 4

Reviewer Name: John Kalbfleisch

Institution and Country: University of Michigan, United States Please state any competing interests or state 'None declared': None declared

Please leave your comments for the authors below See attached file

This is an interesting paper that compares the post-transplant survival outcomes of Black patients in the UK with those of Black patients treated in New York State. These results are surprising and raise a number of questions as to why these geographical differences should occur. The analyses appear to be reasonably well done and there is some good discussion. The paper, however, also contains a great deal of speculation.

Some comments:

It would seem more sensible also to compare White patients in the NY and England. I suspect that there are no large differences between the outcomes for White patients, but that does not seem to be clearly covered in the paper.

Thank you for this comment. As the focus of this paper was on Black kidney transplant recipients, in the interest of space we did not provide any data on differences between White transplant recipients. We have taken the liberty of undertaking this analysis and found a reduced kidney transplant failure/rejection rates for White kidney transplant patients in England versus NYS.

I wondered about the choice of the SPARCS data instead of the SRTR / UNOS data, which would also provide follow up on a national rather than a state wide basis. Can you comment on the comparative accuracy of these data bases and why the choice of SPARCS was made.

Our centre has an institutional agreement to obtain access to SPARCS and therefore was able to analyse the data compared to HES. We agree a more definitive study should access both UNOS/SRTR data from the US and NHSBT Transplant Registry data in the UK for a more in depth analysis but these data resources were not available for analysis at this time.

On Page 9, lines 8 and 9, it is stated the "Recipients identified as having kidney transplant rejection/failure were added as a time variate covariate in the logistic model for mortality." I am not sure what this statement means, but if it is stating that graft failure is included in mortality models as a covariate, then this is not a good thing to do. Graft failure is highly predictive of subsequent mortality and if one population has a higher rate of graft failure, adjusting for tha could remove or create a mortality difference. This needs explanation and may require reanalysis.

Thank you for this comment. We have taken the liberty of re-analysing the data after removing kidney transplant rejection/failure as a time variate factor and this data is now provided. We have also amended our details regarding the statistical analysis in the Methodology.

In the Introduction and elsewhere it is remarked that it is unclear as to whether outcomes of Black patients in one country generalize to another, although this seems clear on general grounds,

especially with the difference between black and white patients in the US, which is not seen elsewhere. It is also remarked that this is the first comparison between countries of posttransplant survival of black patients, yet the Canadian/US study is also referenced, which would seem to contradict the claim.

We have amended our Introduction and now made it clear that this is the first analysis comparing England and patients from the US (page 6).

Covariates collected and used in the analyses do not include some common covariates like BMI or vintage. In addition, there seem to be no covariates on the donor except for living versus deceased. With donors, there are other aspects such as age, cause of death, race, comorbidities etc. that are important in predicting post-transplant outcomes. Were such data unavailable? Is there potential for such covariates to explain at least some of the differences seen. Also, when were the comorbidities determined? Are these comorbidities at baseline (time of transplant) or determined over time prior to transplantation? Some explanation is needed.

We appreciate that this is one of the limitations of this study as we analysed data from administrative data resources rather than recognised transplant registries. This meant many renal and transplant specific factors (e.g. BMI, dialysis vintage, cause of donor death etc) were not available. Cause of death was not available in the SPARCS data and therefore any comparative analysis of this type was not done. This limitation is clearly stated in the Discussion and we hope guides future work in this area using more comprehensive data sets.

The first sentence on page 10 needs revision. This material should refer to table 1. A comparison of demographics for White patients would also be interesting.

The sentence has now been amended. As previously mentioned, the focus of this paper was on Black kidney transplant recipients and therefore space we did not provide any data on differences between White transplant recipients in the interest of space and focus. There is sufficient material for this particular question alone to merit further targeted work specifically focussing on White patients.

Post-transplant outcomes. Are the differences in the relative risks for Whites reported in Table 2 for NY versus England significantly different? Having White as the baseline in Table 2 would seem better to me. This would also lessen the apparent differences in relative risks for Other races.

We are unclear regards to this question. Our analysis is only looking at differences for Black patients between England and NYS.

Top of page 13. Why make this crude inference. Why not just compare NY with England? You could have used SRTR data for a broader comparison.

SRTR data was not accessible to the investigators from outside the US and we have had to utilise SPARCS data. As mentioned in our discussion, a more definitive study should certainly make use of both SRTR and UKTR for US and UK data respectively.

I found the discussion section to be particularly disjointed and unfocused. I think it would be greatly improved with a careful rewriting that reduced its length by about half and avoided repetition of points already made (often several times). A presentation with shorter more focused paragraphs would be much better. Some of the discussion is very speculative – for example the material at the bottom of page 13 and top of page 14 beginning with “This may be related to advances...”. I also found the last paragraph on page 15 confusing. Maybe a section on Study Limitations would help to organize the discussion.

Thank you for this comment and guidance. We have edited the discussion to avoid repetition and

study limitations are covered in the second paragraph of the discussion. We have not made extensive cuts to the discussion in response to the views expressed by other reviewers that the discussion is appropriate and of interest. The current length also accommodates some of the additional comments and discussions that have been requested from other reviewers. We agree much of our discussion is speculative, which we clearly express, but we believe this simply reflects the limited research in this area to support or contradict our findings and we hope this encourages further work in this area.

On page 17, I wonder about the statement “However, after stratification by age and treating kidney transplantation as a competing risk,.....” I’d guess that this analysis is of cumulative incidence and may simply be a result of a higher rate of transplantation among black patients in England. It would be better to look at the rate of failure among patients alive with functioning graft (e.g. a cause specific hazards analysis).

Thank you for this comment. We did not undertake any analysis by stratifying kidney transplantation as a competing risk in our analysis.

VERSION 2 – REVIEW

REVIEWER	Jennifer C Gander Emory University
REVIEW RETURNED	16-Jan-2017

GENERAL COMMENTS	The authors did a nice and thorough job revising the manuscript. The remaining comments I have focus on original comments that were not addressed within the revision. The remaining concern is the lack of information in the Abstract and Participants and Measures. The authors need to update the Introduction with recent USRDS data from the 2016 Annual Report. Other very minor comments are below. I suggest with the updates to the Abstract and Introduction, the editors should accept the manuscript for publication. The reviewer also provided a file in addition to this comment. Please contact the publisher for full details.
--

REVIEWER	Babak Orandi University of California, San Francisco, U.S.A.
REVIEW RETURNED	17-Jan-2017

GENERAL COMMENTS	The authors have satisfied my queries. The sensitivity analyses are reassuring that the conclusions are valid. The only additional things I ask is that the authors 1) put the number of patients missing race information in the body of the manuscript, and 2) for reproducibility, describe how the missing race patients were classified (were they in the "Other" category?).
---

REVIEWER	Natalie Staplin Nuffield Department of Population Health, University of Oxford, United Kingdom
REVIEW RETURNED	23-Jan-2017

GENERAL COMMENTS	The authors have adequately responded to my comments and clarified the statistical methods used. Therefore, I have no further
--

	comments on this manuscript.
--	------------------------------

REVIEWER	John Kalbfleisch University of Michigan United States
REVIEW RETURNED	27-Jan-2017

GENERAL COMMENTS	This paper has been greatly improved by this revision, and provides an interesting contribution to the literature on kidney transplantation. Most of my previous comments and questions were adequately addressed by the authors. I have only two points that I would like to make. 1. In Table 4, it is reported that failure rates are much higher for the post 2007 period than for the pre 2007 period. This seems to contradict many other reports that show improving outcomes over time, at least in the United States. See, for example, the Annual Data Report of the SRTR. This should be checked and, if the result is correct, some comment is needed to understand why such a large difference should be seen. 2. I still think that it would be of interest to include comparisons between the outcomes of whites in NY and the UK. The authors report in their response that such differences exist; there is then the possibility that it is just better outcomes overall in the UK that is being reflected in the racial differences seen. I believe the odds ratios and relative risks of blacks versus whites are not in fact significantly different. The results in the paper are interesting, but an analysis like that presented in Table 4 that looked simultaneously at racial groups and the two countries could be of substantial interest and perhaps give additional insights.
--

VERSION 2 – AUTHOR RESPONSE

Thank you very much for the opportunity to make some minor revisions to our manuscript in light of the recent review. We have made all edits suggested and added the relevant material. Specifically, we have added a supplementary table to show outcomes between white patients in England versus NYS and add some points in our discussion. We have also added a supplementary figure showing study cohort exclusions. Up-to-date USRDS data is quoted and cited, discussion of graft failure rates pre and post 2007 and other queries highlighted in the recent review.

We hope these edits meet with your approval and we look forward to your response in due course.

VERSION 3 – REVIEW

REVIEWER	Babak Orandi University of California San Francisco
REVIEW RETURNED	14-Feb-2017

GENERAL COMMENTS	The authors have satisfied my inquiries. Congratulations to them for a well-done manuscript.
--

REVIEWER	John Kalbfleisch
-----------------	------------------

	University of Michigan United States
REVIEW RETURNED	20-Feb-2017

GENERAL COMMENTS	This is a nice contribution on an important topic.
--